# Assessment of Outdoor Thermal Comfort in Serbia's Urban Environments during Different Seasons

**Milica Lukić** [1,*]**, Dejan Filipović** [1]**, Milica Pecelj** [2,3]**, Ljiljana Crnogorac** [4]**, Bogdan Lukić** [1]**, Lazar Divjak** [4]**, Ana Lukić** [4,5] **and Ana Vučićević** [4,5]

[1]  Faculty of Geography, University of Belgrade, Studentski trg 3/3, 11000 Belgrade, Serbia; dejan.filipovic@gef.bg.ac.rs (D.F.); bogdan.lukic@gef.bg.ac.rs (B.L.)
[2]  Geographical Institute Jovan Cvijić, Serbian Academy of Science and Arts, Đure Jakšića 9, 11000 Belgrade, Serbia; m.pecelj@gi.sanu.ac.rs
[3]  Department of Geography, Faculty of Philosophy, University of East Sarajevo, Alekse Šantića 1, 71420 Pale-East Sarajevo, Republic of Srpska, Bosnia and Herzegovina
[4]  Ph. D. Student, Faculty of Geography, University of Belgrade, Studentski trg 3/3, 11000 Belgrade, Serbia; ljiljanac89@gmail.com (L.C.); Lazar.divjak@undp.org (L.D.); ana.lukic@putevi-srbije.rs (A.L.); ana.vucicevic@putevi-srbije.rs (A.V.)
[5]  Roads of Serbia Public Enterprise, Bulevar kralja Aleksandra 282, 11000 Belgrade, Serbia
*  Correspondence: mlukic@gef.bg.ac.rs; Tel.: +381-653-439-272

**Abstract:** The urban microclimate is gradually changing due to climate change, extreme weather conditions, urbanization, and the heat island effect. In such an altered environment, outdoor thermal comfort can have a strong impact on public health and quality of life in urban areas. In this study, three main urban areas in Serbia were selected: Belgrade (Central Serbia), Novi Sad (Northern Serbia), and Niš (Southern Serbia). The focus was on the temporal assessment of OTC, using the UTCI over a period of 20 years (1999–2018) during different seasons. The main aim is the general estimation of the OTC of Belgrade, Novi Sad, and Niš, in order to gain better insight into the bioclimatic condition, current trends and anomalies that have occurred. The analysis was conducted based on an hourly (7 h, 14 h, and 21 h CET) and "day by day" meteorological data set. Findings show the presence of a growing trend in seasonal UTCI anomalies, especially during summer and spring. In addition, there is a notable increase in the number of days above the defined UTCI thresholds for each season. Average annual UTCIs values also show a positive, rising trend, ranging from 0.50 °C to 1.33 °C. The most significant deviations from the average UTCI values, both seasonal and annual, were recorded in 2000, 2007, 2012, 2015, 2017, and 2018.

**Keywords:** outdoor thermal comfort; UTCI; urban environments; Serbia

## 1. Introduction

The main goal of any prosperous urban community is to create a city of high performance that will be able to meet the needs of all its inhabitants, a city with a preserved environment rich in greenery, and a city that supports public health. All of this ultimately leads to a high standard of urban living. This cannot be expected without paying sufficient attention to the examination of urban outdoor thermal comfort (OTC) [1–4]. The measures to establish a high quality of urban life should also include measures of adequate thermal comfort in urban areas. In light of climate change, the heat load in urban areas has emerged as a serious issue, affecting the well-being of the population and the environment [5]. According to ASHRAE Standard 55 [6] thermal comfort is defined as the "condition of mind that expresses satisfaction with the thermal environment and is assessed by subjective evaluation". So, literally speaking, without satisfaction with the thermal environment, we can hardly imagine adequate urban living conditions. Making a comfortable thermal environment is a strenuous process, from mitigation of the urban heat island effect and

energy saving to public health and welfare as the final goal [7]. Achieving this is very challenging due to many anthropogenic factors that affect urban OTC [5,8–15].

Why is urban OTC so important? First of all, unfavorable thermal comfort and heat stress can negatively affect human health [16] and performance, causing reduced productivity in everyday activities [17]. Certain social groups are more affected by such weather conditions including the elderly, people with chronic illnesses, children, workers in industry, construction, energy-production, etc. [18]. Therefore, the issue of OTC in urban areas has become the subject of the numerous studies over the past two decades [1,2,11,19–29].

The field of human bioclimatology has advanced tremendously [30]. The development path of the indices for assessing OTC has led from simple indicators to advanced and very precise human thermal models (HTMs) [31]. One of the OTC indices used most frequently today is the Universal Thermal Climate Index (UTCI), which belongs to the group of HTMs, and it can be applied globally [32]. As Binarti et al. highlighted, different studies have shown that the multi-node model applying the UTCI is highly accurate in replicating the human dynamic thermal response under a wide range of thermal environments [31]. As Pecelj et al. claim, the UTCI provides many opportunities for researchers from diverse scientific, social, economic, and ecological fields, including weather forecasting for outdoor activities, public health, tourism, industry, ecology, housing, urban planning, sustainable development, etc. [21] (p. 10).

The results of different bioclimatic and biometeorological studies have proven the great applicability of this method in the assessment of urban OTC. For instance, the UTCI has been applied for these purposes worldwide, e.g., Canada [23,33], Brazil [34,35], Australia [32], India [36], China [32,37,38], South Korea [33], Iran [39,40], Saudi Arabia [19], Russia [23], Greece [1,41], Czech Republic [42], Poland [17], France [43], other European regions [20,44,45], etc.

In recent years urban OTC, the application of UTCI, and other thermal indices have become the subjects of many scientific studies by numerous researchers from Serbia and other Western Balkan countries [21,22,25,46–51]. Regarding the application of UTCI in studies of Serbian urban bioclimate, this index was analyzed, e.g., by Pecelj et al. in order to compare bioclimatic conditions in the urban and suburban zones of Belgrade [52]. Belgrade's urban climate in relation to HL and UTCI was once again examined when Pecelj et al. [53] examined biothermal conditions in different geographical environments in Serbia. In addition, a summer UTCI-based assessment of the urban OTC in Belgrade was conducted by Lukić and Milovanović [54]. Bajšanski et al. [55] used UTCI to analyze the effects of urban planning on thermal sensations in Novi Sad. In his Ph.D. thesis, Milošević [56] used PET and UTCI for the assessment of the local climate zone classification system (LCZ) of Novi Sad. Bioclimatic conditions in Novi Sad and Niš during the summer, in order to identify biothermal heat hazards, were the subject of another study conducted by Pecelj et al. [21]. The UTCI was also used for that purpose. The studies of the previously mentioned authors made a big step forward in the development of this scientific field in Serbia, which has led to the need for additional research.

The main objective of this paper is to present a general temporal assessment of urban OTC during different seasons, applying the UTCI universal bioclimatic indicator. Another goal is to examine and show the annual and seasonal differences in three different Serbian cities. The importance and necessity of this type of study is reflected in the fact that we cannot truly and fully understand the impact of OTC on urban life without comprehensive bioclimatic analysis. Such research includes the analysis of several different meteorological parameters through the application of a verified model, which is in this case the UTCI. The study is based on hourly and "day by day" data obtained from three urban synoptic stations over a period of 20 years (1999–2018). The case study covers three cities in Serbia (Belgrade, Novi Sad and Niš), which according to the Köppen-Geiger classification have a moderate climate (Cfa and Cfb) [16,57].

## 2. Materials and Method

This section consists of three different subsections which describe the area of research with the geographical coordinates of synoptic stations (Section 2.1. Study area), the meteorological data used in the study (Section 2.2. Data description), and the method used in the analysis (Section 2.3. Method).

### 2.1. Study Area

The Republic of Serbia straddles the boundary between Central and Southeast Europe, encompassing the southern Pannonian Plain and the central Balkans. The geographical position of Serbia determines the macroclimate (Figure 1). Serbia's climate is temperate continental and mountain. According to Köppen–Geiger climate classification [57,58] Cfa and Cfb climates are predominant in Serbia, which are characterized by a warm, temperate, humid climate with warm summers and peak precipitation during late spring and early summer [21]. Since 1961, climate conditions in Serbia have changed, and this is manifested through a significant increase in temperature and a change in precipitation patterns [59]. Among the Balkan countries, Serbia is one of the most drought-prone areas [60].

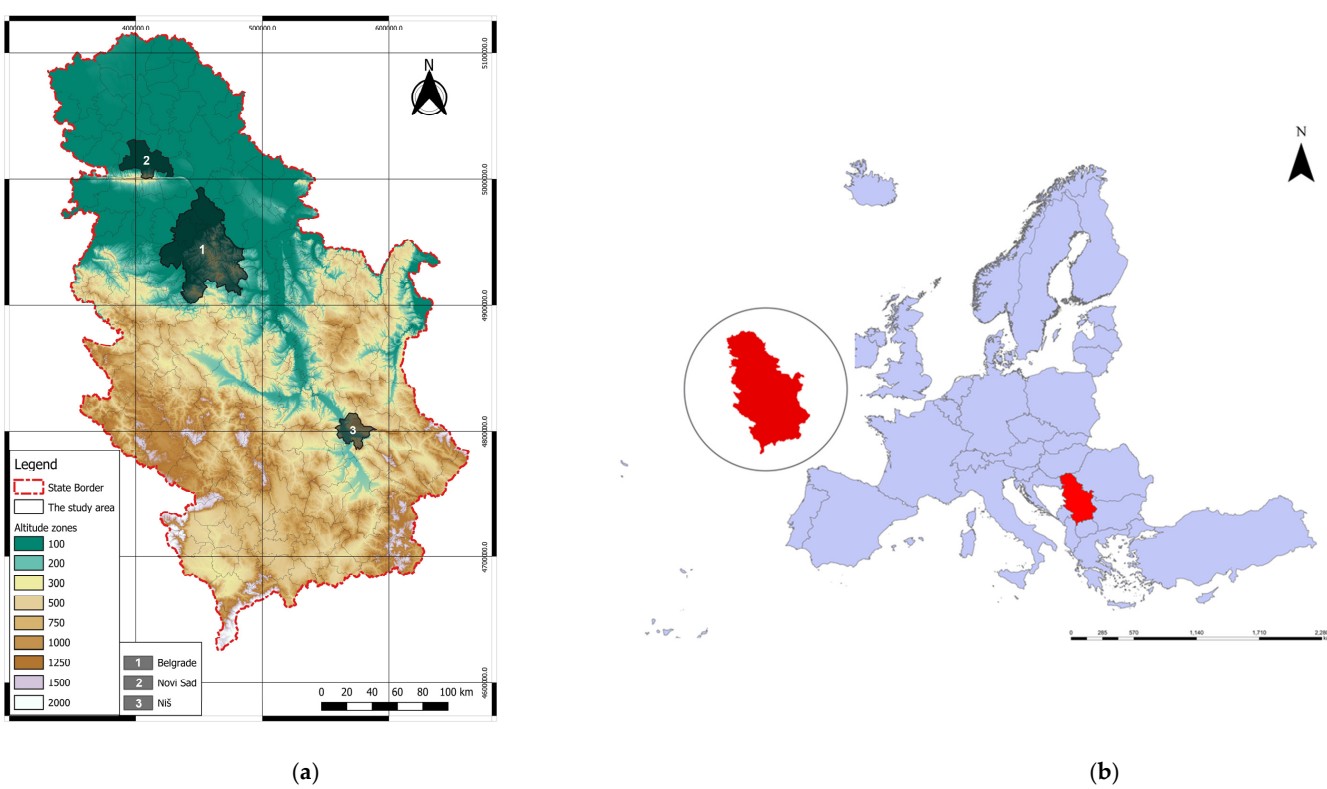

(**a**) (**b**)

**Figure 1.** (**a**) Locations of the studied areas: (1) Belgrade, (2) Novi Sad and (3) Niš; (**b**) Geographical location of Serbia in Europe Figure 1a,b were created using QGIS 3.8 software by OSGeo (Beaverton, OR, USA), based on the Eurostat' official data sets (https://ec.europa.eu/eurostat/web/gisco/geodata/reference-data/ (accessed on 21 May 2021)). Map ratio: 1:1,900,000; projection: WGS 84/UTM, Zone 34N. Altitude structure data of Serbia was obtained by classification of the Digital Elevation Model (DEM) (https://www.opendem.info/ (accessed on 22 May 2021)).

This study covers three synoptic stations located in the three largest and most densely populated urban areas of Serbia: the cities of Belgrade, Novi Sad, and Niš (Figure 1). The first station is the Meteorological Observatory Belgrade (44°48′ N, 20°28′ E, 132 m), located in the central part of the city. Situated in the highly urbanized zone in the most densely populated part of Belgrade, called Vračar, with 19,285 inhabitants/km$^2$ in 2019, it best depicts the outdoor thermal comfort of central urban zones [47,54,61,62]. According to the Köppen-Geiger climate classification, the area of Belgrade belongs to the Cfa type

which is characterized by the humid subtropical climate [54,58]. The mean annual temperature in Belgrade for the 1961–2010 period was 12.3 °C [63], and later that value for the 2000–2017 period was 13.4 °C [64]. The increase in mean daily, monthly and annual temperatures is obvious, and it is additionally affected by the existence of an urban heat island [53,54].

The second meteorological station (45°20′ N, 19°51′ E, 86 m) is located in the district of Novi Sad called Rimski Šančevi, to the north of the central urban zone. This part of the city is mainly used by commercial and industrial activities, and for residential purposes to a lesser extent. Novi Sad is the administrative seat of the Autonomous Province of Vojvodina and is the second-largest metropolitan area of Serbia [13]. Geographically, Novi Sad has a favorable position. The city lies on the banks of the Danube, on the border between the Bačka and Srem regions, facing the northern slopes of Fruška Gora Mountain (Figure 1) [45,49]. The Danube-Tisza-Danube Canal passes through the northern part of the city [16]. According to the Köppen-Geiger classification, Novi Sad has a Cfb climate [58] distinguished by a moderate climate, humid and warm summers. The mean monthly air temperature ranges from −0.3 °C in January to 21.8 °C in July [22].

The third meteorological station is in Niš (43°19′ N, 21°53′ E, 202 m) located in so-called Niš Fortress, in the city center, on the right bank of the Nišava river. Niš is the administrative center and the most significant city of southern Serbia, located in a wide geological depression, the Nišava Basin (Figure 1) [21]. According to Köppen–Geiger classification, this area belongs to the Cfwax type, which represents the Danube type of moderately warm and humid climate. It is characterized by very warm summers (the highest amount of precipitation occurs at the beginning of the summer) and moderately dry winters [21,46,58,65]. The average annual temperature in Niš amounts to 11.8 °C, which is a bit higher than in the other urban areas in this region [65].

### 2.2. Data Description

At the very beginning of the research, an hourly (7 h, 14 h, and 21 h CET) and "day by day" meteorological data sets were collected from a database available from 1999 to 2018 (a period of 20 years). The hourly values of meteorological parameters recorded at 7 h, 14 h, and 21 h provided the best insight into how bioclimatic conditions at one location change throughout the day, from morning to evening. This was the main reason why the authors chose these specific values. The data sets were extracted from the Republic Hydrometeorological Service of Serbia, i.e., from the meteorological yearbooks, for all three synoptic stations [66]. In the calculation process, hourly data on air temperature (t), relative humidity (f), wind speed (v10m) at 10 m above the ground, and cloud cover (cloudiness) were used. Since the Republic Hydrometeorological Service is the reference institution with the full responsibility for the available meteorological data in Serbia, the data sets are considered to be homogeneous, complete, and verified. Two of the three synoptic stations are located in the city center (Belgrade and Niš), while the third (Novi Sad) is located outside of the central urban zone.

### 2.3. Method

This study applies the Universal Thermal Climate Index for the assessment of the outdoor thermal comfort in urban environments.

The UTCI has been made available within the project of the International Society of Biometeorology (ISB) and the framework of the European COST Action 730, as an operational procedure that is used to assess the outdoor thermal environment from the point of view of the core fields of human biometeorology [34]. Since 2009, the UTCI has become one of the most commonly used thermophysiological and bioclimatic indices [17,67]. The UTCI is even recommended by the World Meteorological Organization (WMO) [27]. As Jendritzky et al. claim, the UTCI has unwaveringly acquired the position of an internationally standardized tool for the evaluation of outdoor thermal comfort [68]. Compared to other bioclimatic indices, the UTCI better represents the temporal variability of thermal

conditions. Equally, this index represents various climates, weather, and locations excellently. The UTCI is very sensitive to outdoor factors [69], i.e., changes in air temperature, relative humidity, solar radiation, and wind speed [52]. It is one of the most comprehensive indices for the determination of heat and cold stress in the outdoor environment [70].

As Błażejczyk et al. have defined, the UTCI (°C) is the air temperature of the reference condition that causes the same model response as actual conditions [67] (p. 7). To rephrase it, this model simulates the identical sweat production in the human body response as the genuine environment [17,67]. The UTCI was derived from the Fiala multi-node model, introduced by Fiala et al. [67,69,71]. As this index is an indicator of thermal comfort, it considers both meteorological and physiological parameters describing thermal comfort through the assessment of human energy balance [17,21,67]. As far as physiological conditions are concerned, the metabolic rate (M) has a crucial position. Metabolic processes in the human body produce heat that is continuously exchanged with the surroundings, attaining a state of thermal balance. In that way in the human organism provides a constant body temperature, which is around 37 °C [21,34,67,72]. The amount of heat that is created and released depends on several things, such as physical activity, gender, clothing, age, weight, nutrition, health, different external conditions, etc. [17,21,34,67,72]. The reference environment for this model was defined by the ISB Commission on the UTCI, as [67,69,73,74]:

- A condition of calm air, i.e., wind speed (v10m) 0.5 m/s at 10 m above the ground;
- A mean radiant temperature (Tmrt) equal to air temperature;
- Relative humidity (f) of 50% (capped at 20 hPa for air temperatures over 29 °C).

Physiological parameters (metabolic rate and thermal properties of clothing) are taken as universal constants in the model due to the evaluation by means of regression equation [21]. This implies an outdoor activity where an average person walks at a speed of 4 km/h (1.1 m/s), resulting in a heat production of 135 W/m$^2$ ($\simeq$2.3 MET) of metabolic energy [52,68] and clothing insulation, which is self-adapting according to the environmental conditions [52,75]. It is important to emphasize that clothing insulation, vapor resistance and the insulation of surface air layers are highly affected by changes in wind speed and body movement, which will influence physiological responses [52,69].

Although the application of this model offers numerous advantages to bioclimatic research, researchers have occasionally encountered certain shortcomings in the methodology over the past decade. As Błażejczyk and Kuchcik [76] noted, some of the most common shortcomings of this method include the quality and availability of the meteorological data sets, the method of calculating the mean radiation temperature (Tmrt), determination of wind velocity, etc. In this case, Tmrt was calculated using the BioKlima 2.6 software (Institute of Geography and Spatial Organization, Warsaw, Poland) [77]. BioKlima was developed by Prof. Krzysztof Błażejczyk, as a universal tool for bioclimatic and thermophysiological studies. An integral part of this software package is the model used to determine Tmrt, based on the available data—in this case that were cloudiness (cloud cover, N) and the height of the Sun (Sun altitude, hSl) [76]. Also, some of the authors believe that for the purposes of bioclimatic research in urban areas, it is more relevant to use indices that include wind speed at a height of 1.1 m above the surface (which is equivalent to a standing person's body center) or combine them together with the UTCI [52], as opposed to wind speed at a height of 10 m above the ground, which is the case when we use UTCI [76].

Regarding of thermal stress, this model presents 10 different categories of heat and cold stress, whose description and thresholds are shown in Table 1. The UTCI is calculated as follows:

$$\text{UTCI} = f\ (t, f, v10m, Tmrt) \tag{1}$$

where: t = air temperature (°C), f = relative humidity (%), v10m = wind speed (m/s) at a height of 10 m above the ground, Tmrt = mean radiant temperature (°C).

**Table 1.** The scale of UTCI assessment and physiological responses [67,74].

| UTCI (°C) | Stress Category | Physiological Responses | Abbr. |
|---|---|---|---|
| UTCI > 46 | Extreme heat stress | Increase in rectal temperature time gradient. Steep decrease in total net heat loss. Averaged sweat rate > 650 gh$^{-1}$, steep increase. | EHS |
| 38 < UTCI < 46 | Very strong heat stress | Low core–skin temperature gradient. Increase in rectal temperature at 30 min. | VSHS |
| 32 < UTCI < 38 | Strong heat stress | Averaged sweat rate > 200 gh$^{-1}$. Increase in rectal temperature at 120 min. Instantaneous change in skin temperature. | SHS |
| 26 < UTCI < 32 | Moderate heat stress | Change of slopes in sweat rate and rectal and skin (mean, face, hand) temperature. Occurrence of sweating at 30 min. Steep increase in skin wettedness. | MHS |
| 9 < UTCI < 26 | No thermal stress | Averaged sweat rate > 100 gh$^{-1}$. Plateau in rectal temperature time gradient. | NTS |
| 0 < UTCI < 9 | Slight cold stress | Local minimum of hand skin temperature. | SLCS |
| −13 < UTCI < 0 | Moderate cold stress | Vasoconstriction. Face skin temperature at 30 min < 15 °C (pain). | MCS |
| −27 < UTCI < −13 | Strong cold stress | Numbness. Increase in core–skin temperature gradient. | SCS |
| −40 < UTCI < −27 | Very strong cold stress | Frostbite, numbness, shivering. Steeper decrease in rectal temperature. | VSCS |
| UTCI < −40 | Extreme cold stress | Frostbite. Decrease in rectal temperature time gradient. | ECS |

The basis of any study are quality and complete data sets, so at the very beginning it was necessary to create a database and collect all the meteorological parameters required for the calculation (air temperature, air pressure, relative humidity, wind speed, and cloud cover data). Then, the values of the UTCIs and classification of cold or heat stress were determined using specific software (BioKlima 2.6). After that, the obtained results were presented through morning (UTCI$_{07h}$), midday (UTCI$_{14h}$), and evening (UTCI$_{21h}$) UTCI values, for each weather station. These hourly values of the considered index allow us to monitor how the outdoor thermal comfort of urban zones changes throughout the day, from morning to evening, during each season. Also, as Pecelj et al. pointed out in their study, these hours correspond to the most common time of occurrence of daily minimum and maximum air temperatures [52]. The results were shown by season (spring, summer, autumn and winter) and "year by year" (from 1999 to 2018). At the end of the research the mean seasonal and mean annual values of UTCI at 07 h, 14 h, and 21 h CET in Belgrade, Novi Sad, and Niš were provided, together with the trends for each year. Microsoft Excel 2010 was used to create the figures and tables presented in this paper. The list and definition of UTCIs used in the study are shown in Table 2.

**Table 2.** Definition of UTCIs used in the study.

| Abbreviations | UTCIs | Definition |
|---|---|---|
| UTCI$_{07h}$ | Universal Thermal Climate Index at 07 h | UTCI$_{07h}$ = f (t$_{07h}$, f$_{07h}$, v10m$_{07h}$, Tmrt) |
| UTCI$_{14h}$ | Universal Thermal Climate Index at 14 h | UTCI$_{14h}$ = f (t$_{14h}$, f$_{14h}$, v10m$_{14h}$, Tmrt) |
| UTCI$_{21h}$ | Universal Thermal Climate Index at 21 h | UTCI$_{21h}$ = f (t$_{14h}$, f$_{14h}$, v10m$_{14h}$, Tmrt) |

## 3. Results

This section is divided into the five sub-sections. The first four sub-sections consider the four seasons (spring, summer, autumn and winter). OTC was evaluated through three different values of UTCI: UTCI$_{07h}$ (morning data values at 07:00 CET), UTCI$_{14h}$ (midday data values at 14:00 CET) and UTCI$_{21h}$ (evening data values at 21:00 CET). This applies to all three synoptic stations—Belgrade, Novi Sad and Niš, over the period of 20 years. The results of the OTC assessment are shown in Figures 2–5. The last sub-section is dedicated to the mean seasonal and mean annual values of the UTCIs, which are presented in Tables 3–7 and Figure 6.

### 3.1. Spring OTC in Belgrade, Novi Sad and Niš

In terms of outdoor thermal comfort in big Serbian cities, spring is rated as the most pleasant season for outdoor activities. The prevalent category of OTC during the spring in Belgrade, Novi Sad, and Niš is the one marked as "no thermal stress" (NTS), where the value of UTCI is between 9 °C and 26 °C. The comfortable feeling of being outdoors is primarily present during the early morning (UTCI$_{07h}$). The NTS category accounts for a 65% share in the total number of days, at each of the synoptic stations (Figure 2).

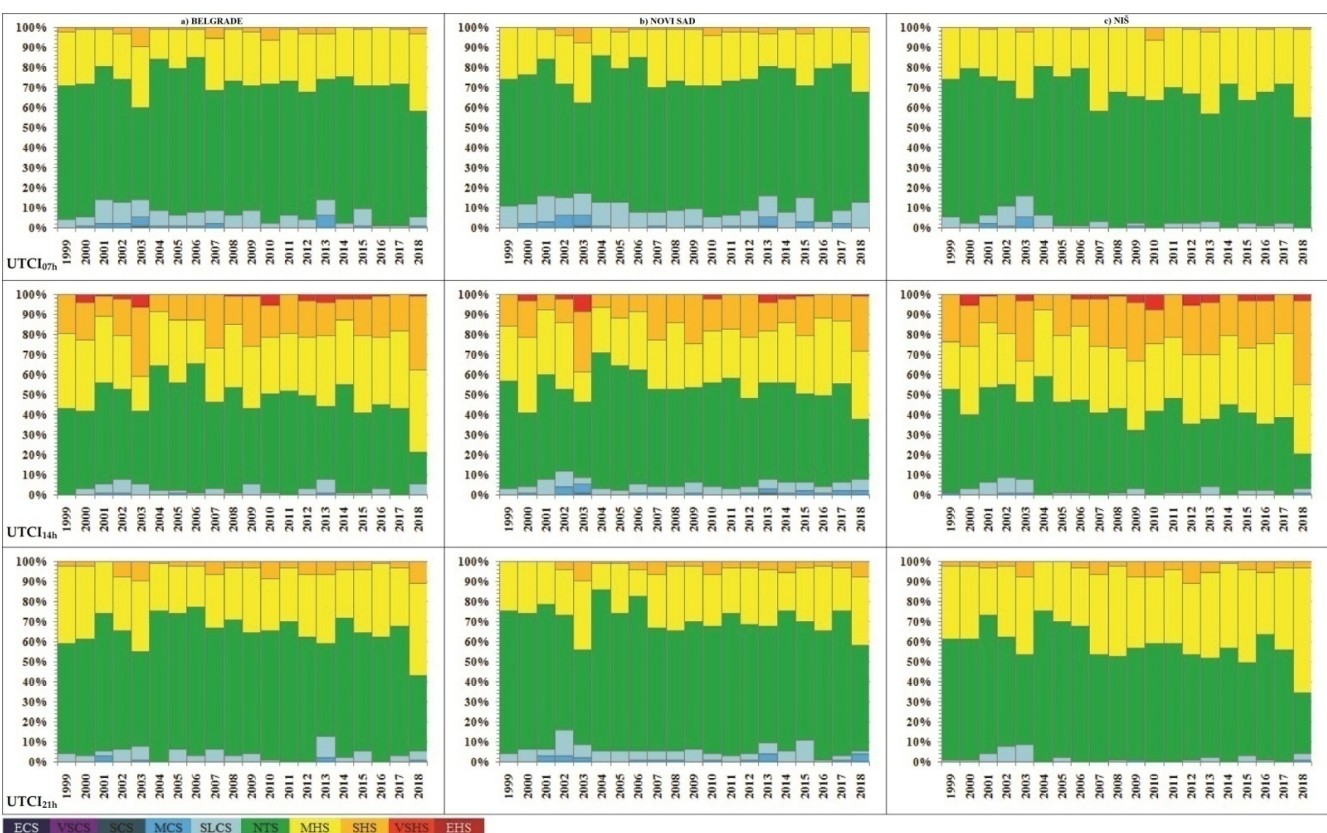

**Figure 2.** Frequencies of different cold and heat stress category in (**a**) Belgrade; (**b**) Novi Sad, and (**c**) Niš based on UTCI$_{07h}$ (first column), UTCI$_{14h}$ (second column), UTCI$_{21h}$ (third column) during spring, over a period of 20 years (1999–2018).

Evenings are somewhat warmer than mornings, and as a result of that the presence of the NTS category is a bit lower (based on comparison of UTCI$_{21h}$ and UTCI$_{07h}$). In Belgrade it accounts for a 61.5%, and 56.67% in Niš' share. On the other hand, the share of moderate heat stress (MHS) and strong heat stress (SHS) is constantly increasing, which can be clearly seen in Figure 2. Moreover, comparing the first (1999–2008) and the second decade (2009–2018) of the investigated period, it was noted that the SHS category in some cases recorded a growth of up to 50%. For instance, during the first decade in Niš there were four days in the SHS category in the morning (07h). During the second decade, 11 such days were observed. The same goes for the UTCI$_{21h}$ in Niš, i.e., from 27 to 49 days in the SHS category. Regarding the UTCI$_{14h}$, the increase in the values is even more obvious. The UTCI$_{14h}$ more frequently exceeded the threshold for the category of very strong heat stress (VSHS) during the spring season. In Niš, the number of days in the VSHS category doubled from 14 days (1999–2008) to 29 such days (2009–2018). The OTC in Novi Sad during the spring months was a bit more favorable compared to Niš, and significant deviations were not recorded in the VSHS category. However, there is a slight but steady increase in the number of days in the SHS category, for all the UTCIs. Belgrade has a very similar outdoor thermal comfort as Niš, and an increase in the number of days was registered in all categories of heat stress (MHS, SHS and VSHS). The continuous increase in

the value of $UTCI_{07h}$, $UTCI_{14h}$ and $UTCI_{21h}$ is evident, and it is expected that this positive trend will continue in the future.

### 3.2. Summer OTC in Belgrade, Novi Sad and Niš

Summer is the least suitable season in terms of bioclimatic conditions in urban areas, with a high level of heat load, high air temperatures, and high probability for heatwave occurrence. According to the obtained results for the summer season, 2000, 2007, 2012, 2015, and 2017 stand out as the years with the most severe urban OTC (Figure 3).

In the morning OTC in Belgrade, Novi Sad, and Niš is slightly more comfortable than later during the day. Mornings ($UTCI_{07h}$) are characterized by MHS as a dominant category of thermal stress. In particular, physiological stress in the category of "strong heat stress" (SHS) and "very strong heat stress" (VSHS) occurred quite frequently during the investigated period. Comparing the two decades of the research period 1999–2008 and 2009–2018, it was found that the number of days when NTS occurred decrease, while on the other hand the number of days with a higher level of heat load increased. This applies to all three synoptic stations, and to all three UTCIs. The VSHS in the evening ($UTCI_{21h}$) during this twenty-year period was registered several times in Belgrade and Novi Sad. It happened in Belgrade in 2007, 2013 and 2015, and in Novi Sad in 2000 and 2017. The maximum value of $UTCI_{21h}$ in Belgrade was 39.11 °C (on 29 July 2013), while in Novi Sad maximum value of $UTCI_{21h}$ was measured on 5 August 2017, and it was 38.17 °C. These events were connected to the occurrence of tropical nights (Tmin > 20 °C).

Generally, physiological stress during summer months in these cities is described as strong or very strong, especially when we consider $UTCI_{14h}$ and this index best reflects the bioclimatic conditions that occur in the hottest part of the day. As we can see on the graphs, Niš stands out in terms of thermal discomfort. Due to its geographical position and the continentality of the region, a larger number of heatwaves have been recorded in Niš (both during winter and summer) compared to Belgrade [78] and especially to Novi Sad. Days with SHS, VSHS, and EHS in Novi Sad together account for over 53% of the total number of days. In Belgrade, this share is 57.61%, and in Niš it reaches 66.86%.

Extreme weather conditions in the Balkan region are more common in the hot part of the year than in the cold season, so extreme OTC usually occurs in the summer. An unprecedented heatwave hit almost the entire territory of Serbia in 2007, and it lasted from 14 to 24 July [79]. It was then that the highest values of $UTCI_{14h}$ during this twenty-year period were recorded. On 24 July 2007, the $UTCI_{14h}$ exceeded the threshold value for extreme heat stress (EHS) in all three cities. In Belgrade the value of $UTCI_{14h}$ was 48.1 °C, while in Novi Sad it was 47.12 °C. For these two cities, this was the only case when thermal comfort was assessed as extreme. As Tošić et al. have highlighted [64], on the same date an air temperature of 43.6 °C was measured, which was the highest recorded temperature in Belgrade in the past 120 years. Moreover, the highest increase in the maximum temperature, 3.1 °C, which previously dated back to 1888, was registered on that summer in Belgrade [79].

In terms of EHS, Niš is in the first place, with six such days. The highest value of $UTCI_{14h}$ in Niš was also measured on 24 July, and was 48.73 °C. It is important to note that on 24 July 2007, a temperature of 44.9 °C was registered in Smederevska Palanka, which was the absolute maximum value ever recorded in Serbia [75,80]. The remaining five times when the EHS was recorded in Niš were on 5 July 2000 ($UTCI_{14h} = 47.03$ °C), then on 20 July 2007 ($UTCI_{14h} = 46.41$ °C), on 23 August 2007 ($UTCI_{14h} = 46.17$ °C), on 5 August 2017 ($UTCI_{14h} = 46.78$ °C), and the last one was on 6 August 2017 ($UTCI_{14h} = 46.81$ °C). The OTC in Niš is generally more severe compared to the other two stations, especially during the summer.

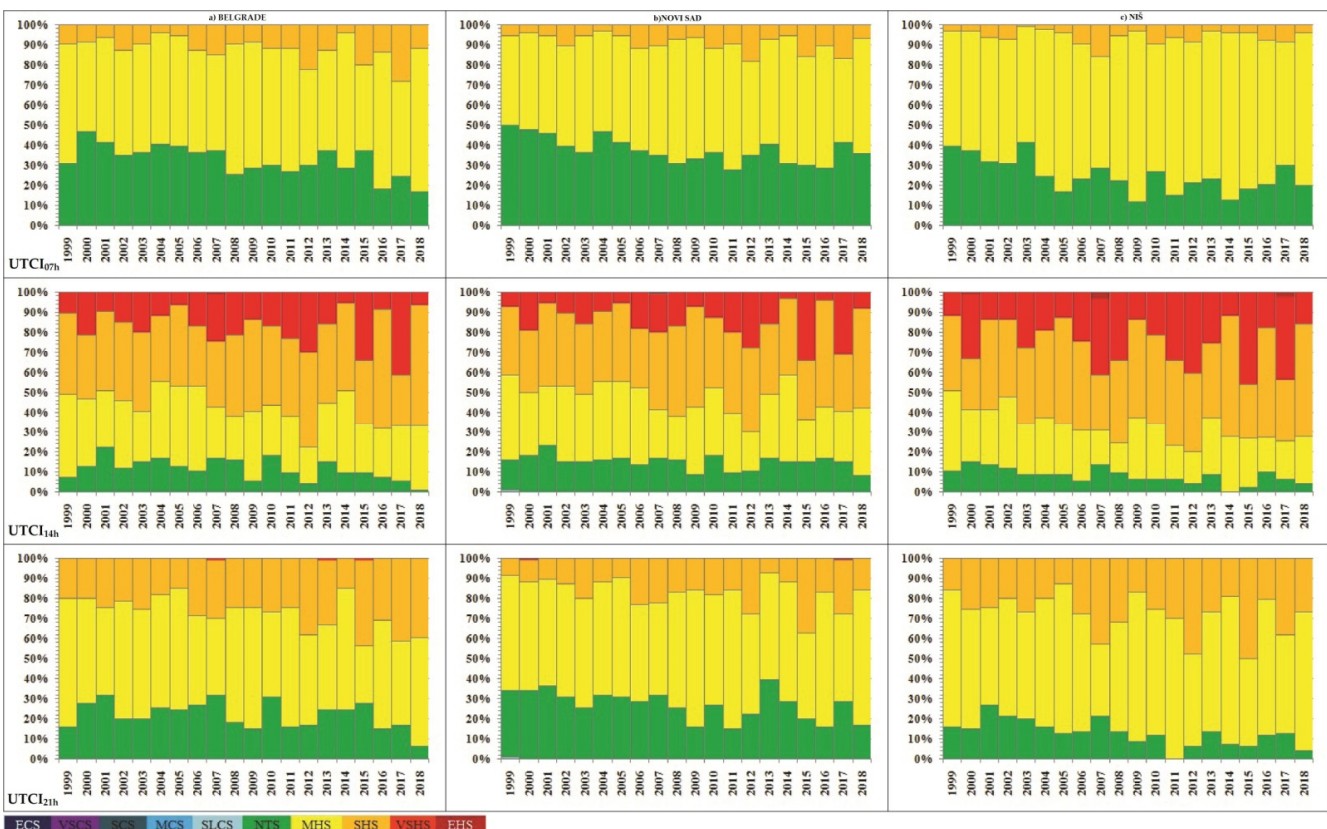

**Figure 3.** Frequencies of different cold and heat stress category in (**a**) Belgrade; (**b**) Novi Sad; and (**c**) Niš based on UTCI$_{07h}$ (first column), UTCI$_{14h}$ (second column), UTCI$_{21h}$ (third column) during summer, over a period of 20 years (1999–2018).

### 3.3. Autumn OTC in Belgrade, Novi Sad and Niš

In general, autumn is the most suitable part of the year after spring, in terms of work, passive and active recreation, and other types of physical outdoor activities. Based on the obtained results, there is a clear decrease in the number of days during the autumn when UTCIs belonged to one of the categories of cold stress (Figure 4). Important changes are related to the category of strong cold stress (SCS). For instance, morning SCS in Niš occurred six times during the first decade (1999–2008), and not even once during the second decade (2009–2018). The same goes for midday and evening UTCI values. During the first decade there were seven days (UTCI$_{14h}$) and five days (UTCI$_{21h}$) with SCS, and during the second ten years, there was not a single one. As for the UTCI$_{07h}$, Belgrade has a similar trend. During the first ten years of investigated period there were nine days with SCS, and only two in the second decade, in 2009 and in 2014.

On the other hand, the average value of the UTCI$_{07h}$, UTCI$_{14h}$, and UTCI$_{21h}$ constantly increased, so the days without thermal stress were increasingly registered during autumn. The same goes for the category of moderate heat stress (MHS), which occurred more frequently in recent years. Evening values of UTCI in urban areas (UTCI$_{21h}$) are especially interesting; the presence of MHS is slowly becoming more pronounced. The highest level of thermal stress during autumn, for all stations, was VSHS. In Belgrade and Novi Sad, there was only one day when UTCI$_{14h}$ reached the threshold for VSHS. In Belgrade that was on the 26 September 1999 (UTCI$_{14h}$ = 38.20 °C). In Novi Sad that was on the 27 September 2012 (UTCI$_{14h}$ = 38.19 °C). There were five such days in Niš, of which three consecutive were days from 29 September to 1 October 2012. The maximum value of autumn UTCI$_{14h}$ was 39.53 °C (Niš, 30 September 2012).

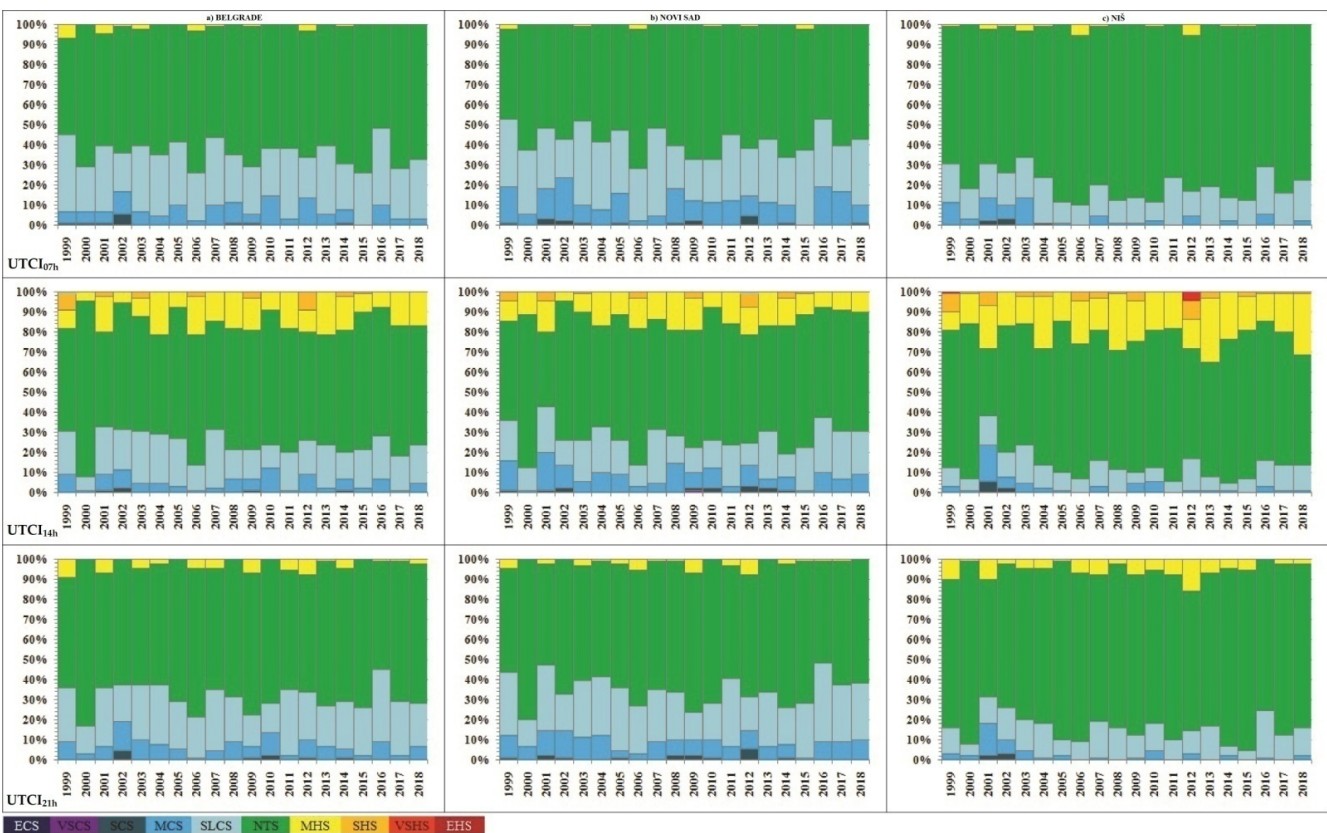

**Figure 4.** Frequencies of different cold and heat stress category in (**a**) Belgrade; (**b**) Novi Sad and (**c**) Niš based on $UTCI_{07h}$ (first column), $UTCI_{14h}$ (second column), $UTCI_{21h}$ (third column) during autumn, over a period of 20 years (1999–2018).

### 3.4. Winter OTC in Belgrade, Novi Sad and Niš

Based on the obtained results shown on Figure 5, we can easily conclude that winters in Niš are the most bioclimatically favorable in comparison to the other two stations. In the morning ($UTCI_{07h}$), the number of days with NTS is almost equal to the number of days in which some of the categories of cold stress occur. However, $UTCI_{14h}$ and $UTCI_{21h}$ show somewhat different bioclimatic conditions. Midday OTC is a more pleasant for outdoor activities, because the predominant category of thermal stress is where the value of $UTCI_{14h}$ is between 9 °C and 26 °C (NTS), with a 61.85% share in the total number of days. There were 11 days when the value of $UTCI_{14h}$ exceeded the threshold for MHS during the winter. The $UTCI_{21h}$ results in Niš show a very similar situation. The dominant category is NTS, with a 59.9% share. In Novi Sad, the highest number of days was recorded in all categories of cold stress, regardless of the examined UTCIs. Dominant categories of physiological stress in this city are slight cold stress (SLCS) and moderate cold stress (MCS). In Belgrade, those categories are NTS and SCS.

Regarding cold stress, the lowest observed category of physiological stress in these urban areas was very strong cold stress (VSCS). It occurs more often in Novi Sad, and least often in Niš. The minimum value was recorded on 31 January 2014 in Belgrade, when the $UTCI_{07h}$ was −34.73 °C. In general, the lowest values are recorded more often in the evening than in the morning, except in special cases. The minimum value in Novi Sad was recorded on the same date as in Belgrade, on 31 January 2014, and $UTCI_{21h}$ was −33.81 °C. The minimum value in Niš was registered on 6 January 2017, when $UTCI_{07h}$ was −30.39 °C. A comparison of the studied meteorological data shows that the appearance of VSCS is connected to the low air temperatures and days with high wind-speeds. It should be pointed out that in the course of the research, a slight increase in daily and hourly winter air temperatures was noted. This also resulted in a gradual increase in the values of the investigated UTCIs.

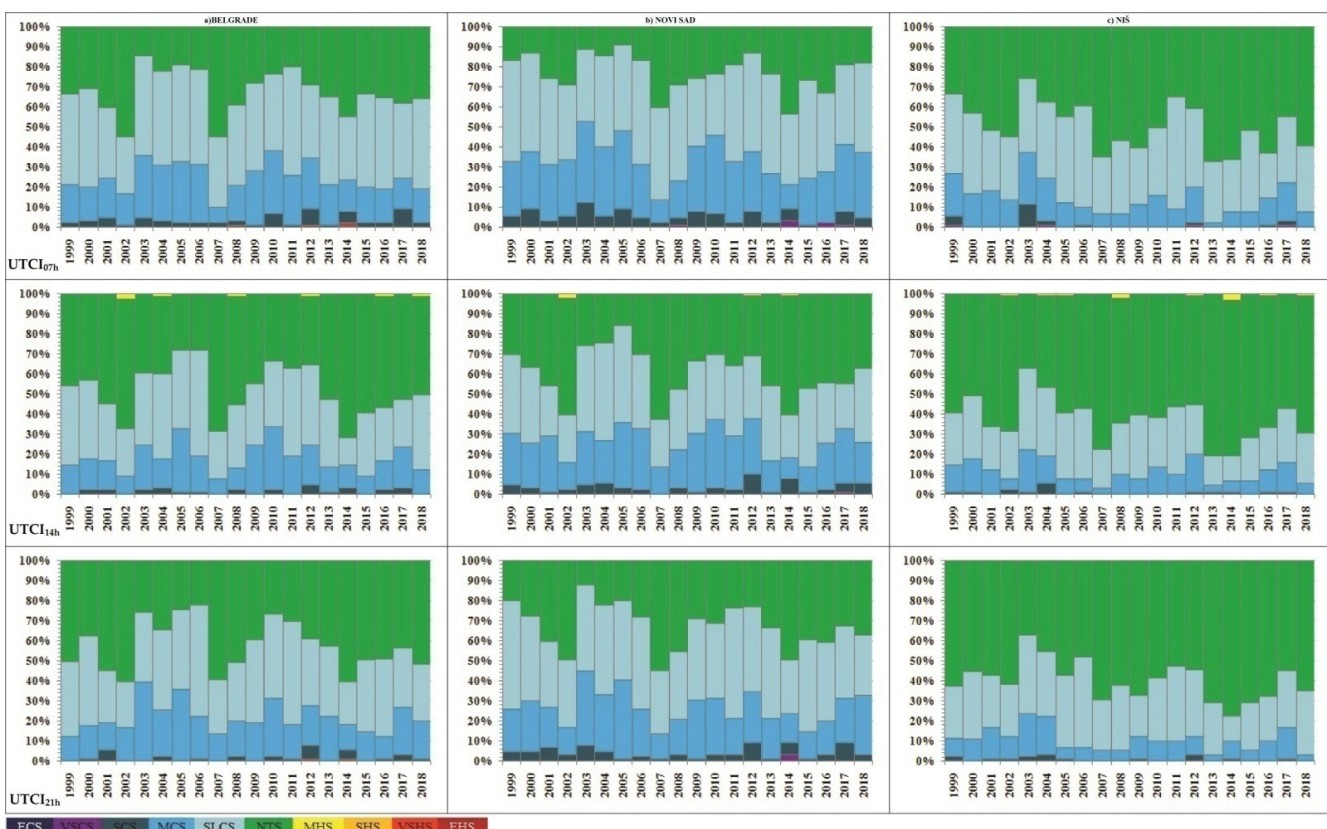

**Figure 5.** Frequencies of different cold and heat stress category in (**a**) Belgrade; (**b**) Novi Sad; and (**c**) Niš based on UTCI$_{07h}$ (first column), UTCI$_{14h}$ (second column), UTCI$_{21h}$ (third column) during winter, over a period of 20 years (1999–2018).

### 3.5. Average Seasonal and Annual UTCIs in Belgrade, Novi Sad and Niš

This subsection presents the average spring, summer, autumn, and winter UTCI values at 07 h, 14 h, and 21 h CET for the 1999–2018 period. In addition, the average decade-long values for the first (1999–2008) and second (2009–2018) decades are also presented. Table 3 presents the average spring UTCI in Belgrade, Novi Sad, and Niš. The average spring UTCI value for Belgrade and Novi Sad is below 26 °C, which means that it belongs to the NTS category. The exception is Niš (UTCI$_{14h}$), where the average thermal stress belongs to the MHS category.

During the calculation of the UTCI, it was noted that the average annual and average seasonal air temperature increased over the years, which resulted in the growth of the index value. As a result, there was an increase in the mean spring UTCI when comparing the two decades of the study period. This difference in some cases exceeds 1 °C. This is especially pronounced in Niš, where the average decade difference is 1.33 °C for UTCI$_{07h}$, 1.49 °C for UTCI$_{14h}$ and 1.2 °C for UTCI$_{21h}$. The average decade-long difference in UTCIs values for Belgrade and Novi Sad is regularly between 0.75 °C and 1 °C. The maximum mean spring UTCI value for all three stations was measured in 2018. This refers to UTCI$_{14h}$, when the average spring index value in Belgrade was 28.32 °C (MHS), in Novi Sad it was 26.19 °C (MHS), and in Niš it was 29.11 °C (MHS).

**Table 3.** Average spring UTCIs values in Belgrade, Novi Sad, and Niš for the period 1999–2018, and average decade-long UTCIs values for the 1st decade (1999–2008) and 2nd decade (2009–2018).

| Spring UTCIs | Belgrade | | | Novi Sad | | | Niš | | |
|---|---|---|---|---|---|---|---|---|---|
| | Average Value | 1st Decade | 2nd Decade | Average Value | 1st Decade | 2nd Decade | Average Value | 1st Decade | 2nd Decade |
| $UTCI_{07h}$ | 21.07 | 20.59 | 21.56 | 19.98 | 19.60 | 20.35 | 22.38 | 21.71 | 23.04 |
| $UTCI_{14h}$ | 25.51 | 25.01 | 26.01 | 24.10 | 23.70 | 24.50 | 26.49 | 25.74 | 27.23 |
| $UTCI_{21h}$ | 23.24 | 23.22 | 23.27 | 21.51 | 21.01 | 22.01 | 24.17 | 23.57 | 24.77 |

According to the obtained results (Table 4), the average summer UTCI falls within two categories of thermal stress: MHS and SHS. Compared to the previous case, a slightly lower difference between the average values of the index was observed. For summer season this difference is between 0.54 °C and 1.24 °C. The biggest difference in terms of changes at the level of two decades occurred in Belgrade. The difference in the average summer UTCI between the first and second decade in Belgrade is 1.05 °C for $UTCI_{07h}$, 1.24 °C for $UTCI_{14h}$, and 0.97 °C for $UTCI_{21h}$. On the other hand, for Novi Sad that difference is between 0.81 °C ($UTCI_{07h}$) and 0.96 °C ($UTCI_{14h}$). In Niš, there is also an increase in values, but it is not as pronounced as during the spring months. In spite of that, we should not neglect it because Niš is certainly characterized by a warmer climate, with hotter and drier summers than the other two cities. Compared to Belgrade and Novi Sad, the highest average summer value of UTCI was recorded in Niš, in the morning, midday, and evening. The maximum mean summer UTCI value for Novi Sad and Niš was measured in 2012—34.04 °C in Novi Sad and 36.34 °C in Niš—while in Belgrade it was for the summer 2017, with 35.27 °C.

**Table 4.** Average summer UTCIs values in Belgrade, Novi Sad, and Niš for the period 1999–2018, and average decade-long UTCIs values for the 1st decade (1999–2008) and 2nd decade (2009–2018).

| Summer UTCIs | Belgrade | | | Novi Sad | | | Niš | | |
|---|---|---|---|---|---|---|---|---|---|
| | Average Value | 1st Decade | 2nd Decade | Average Value | 1st Decade | 2nd Decade | Average Value | 1st Decade | 2nd Decade |
| $UTCI_{07h}$ | 27.40 | 26.88 | 27.93 | 26.80 | 26.39 | 27.20 | 27.81 | 27.45 | 28.16 |
| $UTCI_{14h}$ | 32.75 | 32.13 | 33.37 | 32.02 | 31.54 | 32.50 | 33.76 | 33.49 | 34.03 |
| $UTCI_{21h}$ | 29.15 | 28.66 | 29.63 | 28.15 | 27.71 | 28.59 | 29.83 | 29.31 | 30.35 |

The mean autumn UTCIs for all three observation times are given in Table 5. The results for the autumn months also show a positive trend. Once again, the average values were higher during the 2009–2018 period, compared to the 1999–2008 period. That difference is between 0.13 °C and 1.26 °C, which depends on the part of the day when the measurements were carried out. During the research, a rising trend in air temperature was also noted at the end of September and during October, which affected the increase in the value of autumn UTCI. Compared to the spring season, it was a slight, but still noteworthy increase in UTCI value. The highest average values of autumn UTCI during this twenty-year period were measured in Niš. In addition, the average $UTCI_{14h}$ in Niš during the 1999–2008 period was 18.2 °C, while during the 2009–2018 period it was 19.46 °C. A similar situation was observed in Belgrade: the average $UTCI_{14h}$ for the 1999–2008 period was 15.71 °C, while for the 2009–2018 period it was 16.30 °C.

The maximum mean autumn UTCI value for all three stations was measured in 2006, and this refers to $UTCI_{14h}$. In Belgrade it was 18.64 °C, in Novi Sad it was 18.10 °C, and in Niš it was 20.95 °C.

**Table 5.** Average autumn UTCIs values in Belgrade, Novi Sad, and Niš for the period 1999–2018, and average decade-long UTCIs values for the 1st decade (1999–2008) and 2nd decade (2009–2018).

| Autumn UTCIs | Belgrade | | | Novi Sad | | | Niš | | |
|---|---|---|---|---|---|---|---|---|---|
| | Average Value | 1st Decade | 2nd Decade | Average Value | 1st Decade | 2nd Decade | Average Value | 1st Decade | 2nd Decade |
| $UTCI_{07h}$ | 11.74 | 11.50 | 11.97 | 10.18 | 10.12 | 10.25 | 14.41 | 13.98 | 14.85 |
| $UTCI_{14h}$ | 16.01 | 15.71 | 16.30 | 14.67 | 14.56 | 14.78 | 18.83 | 18.20 | 19.46 |
| $UTCI_{21h}$ | 13.03 | 12.87 | 13.18 | 11.95 | 11.85 | 12.06 | 16.06 | 15.75 | 16.36 |

During the 20-year study period, the average values of winter UTCI in Belgrade and Novi Sad were in most cases below 9 °C, which resulted in the dominance of the SLCS category at the seasonal level. As for the winter UTCI in Niš, the values are slightly higher, which results in a higher presence of the NTS category. Average winter $UTCI_{14h}$ for the 1999–2018 period in Niš was 10.78 °C, while in Belgrade it was 7.97 °C. Novi Sad was in the last place with the lowest value of the average winter $UTCI_{14h}$ (5.89 °C). The highest increase in the average winter index value was also recorded in Niš, especially when it comes to $UTCI_{07h}$. The difference in average values in the morning between the first and second ten-year period was 1.69 °C. Regarding winter $UTCI_{21h}$ in Niš, the difference between the 1999–2008 period and the 2009–2018 period was 1.19 °C (Table 6). A positive trend was also recorded in Novi Sad, but the differences in the average decade values are significantly lower. With regard to the $UTCI_{07h}$, the results show a difference of 0.40 °C, comparing the first and second decades. In terms of $UTCI_{21h}$ this difference is only 0.08 °C.

**Table 6.** Average winter UTCIs values in Belgrade, Novi Sad, and Niš for the period 1999–2018, and average decade-long UTCIs values for the 1st decade (1999–2008) and 2nd decade (2009–2018).

| Winter UTCIs | Belgrade | | | Novi Sad | | | Niš | | |
|---|---|---|---|---|---|---|---|---|---|
| | Average Value | 1st Decade | 2nd Decade | Average Value | 1st Decade | 2nd Decade | Average Value | 1st Decade | 2nd Decade |
| $UTCI_{07h}$ | 4.72 | 4.89 | 4.56 | 2.41 | 2.21 | 2.61 | 7.77 | 6.93 | 8.62 |
| $UTCI_{14h}$ | 7.97 | 8.09 | 7.85 | 5.89 | 5.82 | 5.96 | 10.78 | 10.16 | 11.40 |
| $UTCI_{21h}$ | 6.43 | 6.62 | 6.25 | 4.42 | 4.38 | 4.46 | 9.66 | 9.06 | 10.25 |

The positive trend of average annual UTCI values for each of these weather stations is even more obvious if we look at Table 7 and Figure 6. Table 7 shows the average value of the UTCIs for the 20-year period, as well as the average decade-long values for the first (1999–2008) and second (2009–2018) decades of the investigated period. Looking at the two decades separately, it is clear that there has been an increase in UTCI values due to changes in microclimatic factors (primarily rising air temperatures). The difference in these values is from 0.50 °C to 1.33 °C. The highest changes were recorded at the synoptic station in the center of Niš, where the difference between the mean decade values is 1.27 °C for $UTCI_{14h}$ and 1.33 °C for $UTCI_{21h}$.

**Table 7.** Average annual UTCIs values in Belgrade, Novi Sad, and Niš for the period 1999–2018, and average annual decade-long UTCIs values for the 1st decade (1999–2008) and 2nd decade (2009–2018).

| Ann. UTCIs | Belgrade | | | Novi Sad | | | Niš | | |
|---|---|---|---|---|---|---|---|---|---|
| | Avg. Ann. Value | Avg. 1st Dec. | Avg. 2nd Dec. | Avg. Ann. Value | Avg. 1st Dec. | Avg. 2nd Dec. | Avg. Ann. Value | Avg. 1st Dec. | Avg. 2nd Dec. |
| Ann. $UTCI_{07h}$ | 16.43 | 16.16 | 16.70 | 15.04 | 14.79 | 15.30 | 18.17 | 17.50 | 18.83 |
| Ann. $UTCI_{14h}$ | 20.77 | 20.44 | 21.10 | 19.38 | 19.12 | 19.64 | 22.73 | 22.09 | 23.36 |
| Ann. $UTCI_{21h}$ | 18.04 | 17.79 | 18.29 | 16.70 | 16.44 | 16.97 | 20.13 | 19.59 | 20.68 |

Figure 6 shows the annual value of UTCI at 07 h, 14 h, and 21 h in Belgrade, Novi Sad, and Niš together with trends for each year. The maximum average annual UTCI value at 14 h was registered in Niš, in 2018 (24.01 °C). In the same year, Belgrade registered its maximum at 14 h (22.31 °C). The maximum average annual UTCI at 14 h in Novi Sad was registered in 2015 (21.04 °C). High average annual values were also recorded during 2017, 2014, 2012, and 2007. The maximum average annual value at 21 h was registered in Niš, in 2015 (21.17 °C). For Belgrade it was 19.59 °C in 2018, and for Novi Sad the maximum average annual UTCI at 21 h was 18 °C in 2015. The maximum values of the index measured in the morning are slightly lower compared to the evening values.

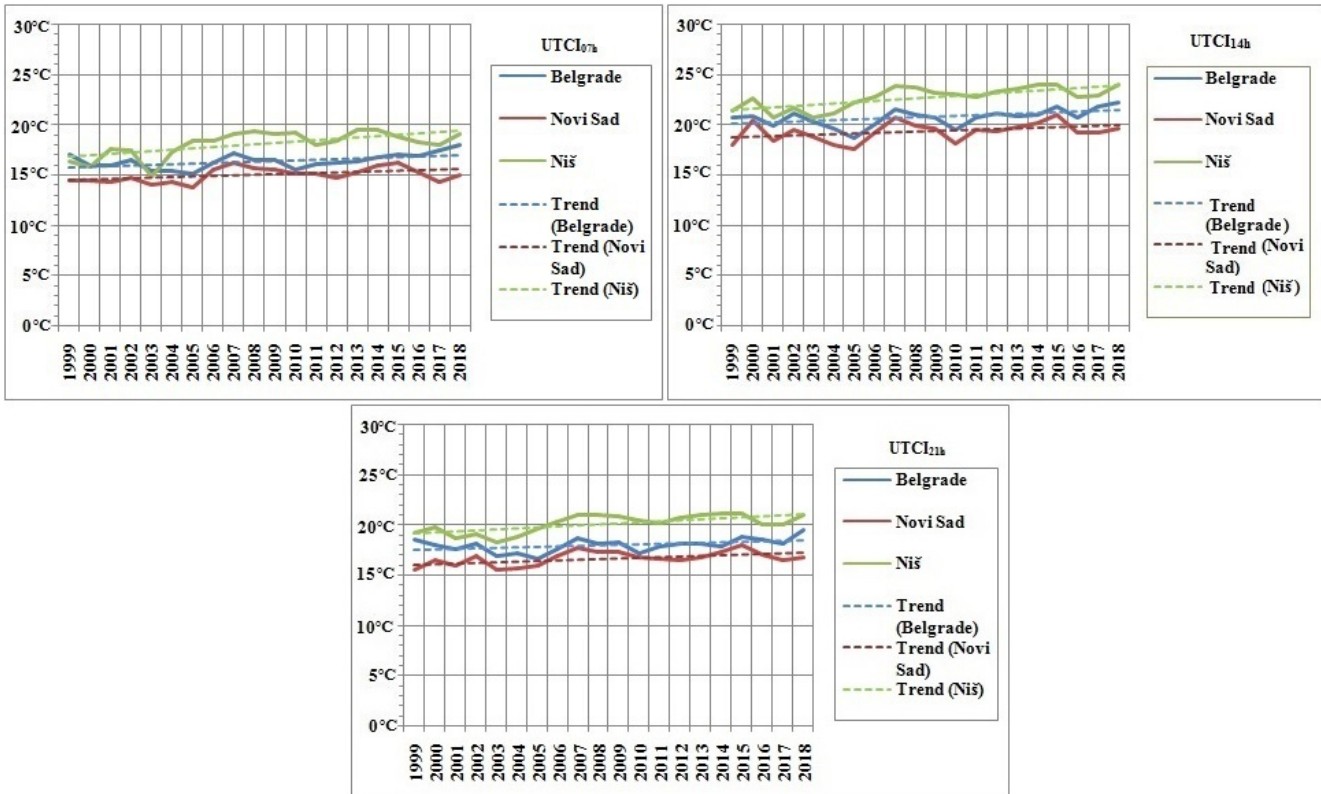

**Figure 6.** Average annual UTCI at 7 h, 14 h and 21 h CET during the period 1999–2018. Blue indicates UTCI values and trends for Belgrade, red is used for Novi Sad, and green for Niš.

## 4. Discussion

This paper presents a temporal assessment of urban outdoor thermal comfort based on the UTCI heat budget index. Meteorological data were obtained from three synoptic stations in three different urban areas in Serbia. Two weather stations (Belgrade and Niš) are located in the central part of the city, while the third one (Rimski Šančevi, in Novi Sad) is located in the vicinity of the central urban zone (7 km north). Biothermal analysis for a period of 20 years (1999–2018) was conducted on the basis of data on air temperature, relative humidity, wind speed, and cloud cover. For this analysis, the morning (07 h), midday (14 h), and evening (21 h) values of meteorological parameters were considered. Based on this, the average seasonal and annual values of the UTCIs were determined, as well as the trends registered during these 20 years.

The impact of climate change on the urban climate has become increasingly apparent in the past several decades. These changes have not bypassed Serbia, and the country has been experiencing a warming trend since 1980 [78]. The negative effect of global warming, in combination with the changed distribution of precipitation, has resulted in prolonged periods of drought and strong precipitation events. Furthermore, there is the increase in frequency and intensity of heatwaves, floods, forest fires, and recently in a disturbance of

the ecological balance [59,80,81]. All of this has a huge effect on the quality of urban life in Serbia.

The findings of this study are in conformity with previously conducted climate research of Belgrade, Novi Sad, and Niš. The most prominent extreme weather conditions were recorded during the summer season. The results of the midday UTCI at 14 h show a significant occurrence of the thermal heat stress (VSHS and ESH) during the summer months. The occurrence of extreme OTC is most evident in the number of days with VSHS. This category of thermal stress is most common during the summer. EHS, which was recorded during the summer during the 1999–2018 period in Belgrade (one day), Novi Sad (one day) and Niš (six days), causes great biothermal discomfort. ESH was always preceded by several days with VSHS. Such conditions can have a strong impact on public health in big cities, and cause health problems such as fatigue, dizziness, headache, sunstroke, heatstroke [46], even sometimes resulting in short-term mortality increases. For instance, July 2007 is remembered as the period of the most severe heat event during this 20-year period. Between 16 and 24 July 2007, there was a total of 167 excess deaths (38%) in Belgrade [82]. This heat wave peaked on 24 July 2007. On that day, the absolute temperature maximum of 43.6 °C was recorded, and it was the only day with EHS, with a daily maximum mortality count of 94 deaths in Belgrade [83]. On the same day, the highest ever recorded temperature in Niš was observed, 44.2 °C [84]. Furthermore, the maximum value of $UTCI_{14h}$ was measured in each of these cities: 48.1 °C in Belgrade, 47.12 °C Novi Sad, and 48.73 °C in Niš. Based on the results of this and other studies, it is presumed that these maximum values will be exceeded at some point in the future. For example, the findings of Unkašević et al. [85] have shown that the mean summer temperature at Belgrade increases at the rate of 0.1316 °C/year. The increase in summer temperature is also proved in this paper. Table 4 clearly shows that there has been an increase in average summer UTCIs values, if we compare the first (1999–2008) and second (2009–2018) decades of the research period. This increase is between 0.54 °C and 1.24 °C. As Bajat et al. [86] have noted, the summer is the season with the largest contribution to annual positive trends.

In this study, Niš stood out as the urban area in Serbia with the greatest thermal discomfort, especially during the summer. The weather in this city can be extremely hot and dry. This is also confirmed by the findings of Tošić and Unkašević [87].

Summer anomalies and severe bioclimatic conditions were also observed in 2000, 2007, 2012, 2015, 2017, and 2018. This is in correlation with other studies that have examined heat waves, climate extremes, droughts, and other thermal hazards [88,89]. The year 2012 was the second hottest year in Serbia since 1951, with the greatest number of tropical days [66]. The summer of 2015 is remembered as one of the hottest summers in the past several decades, with extreme temperatures and prolonged heat waves not just in Serbia, but also in Central Europe [16].

Although the most striking extreme bioclimatic conditions were recorded during the summer of 1999–2018, the importance of the positive trends recorded during the spring and autumn months should not be underestimated. According to the obtained results, there is a clear decrease in the number of days in which spring and autumn UTCIs belonged to one of the categories of cold stress. The most pronounced decline of CS was recorded in Belgrade and Niš. On the other hand, it was noted that the SHS category in some cases recorded a growth of up to 50% in the spring. Lazić et al. [90] have shown that the mean annual temperature measured at the Rimski Šančevi station (Novi Sad) had a rising trend of 0.024 °C/decade over a period of 40 years (1951–1990). For springtime that trend was 0.244 °C/decade [90]. Today that value is even higher. In Belgrade, the situation is similar. For example, the mean spring temperature in Belgrade has a positive trend of 1.32 °C/100 years, while the minimum spring temperature records has positive trend of 1.92 °C/100 years. For $T_{max}$ that trend is 0.54 °C/100 years [91]. This increase in temperature results in an increase in the index value, which is evident in Table 3, which shows the average spring UTCI at 07 h, 14 h, and 21 h CET, as well as the average decade-long UTCIs. This is especially pronounced in Niš, where the average decade difference

for $UTCI_{14h}$ goes up to 1.49 °C (from 25.74 °C during the 1999–2008 period to 27.23 °C during the 2009–2018 period). There is no mistake in claiming that spring days in Serbia became warmer.

The results obtained in this study for the 1999–2018 autumn period show also a rising UTCI trend. Unkašević and Tošić [78] noted that for decades there has been a negative trend in the number of cold days and a positive trend in the number of warm days and nights—both indicating warming in Serbia. Figure 4 shows that the days with NTS in Belgrade, Novi Sad, and Niš are registered more often during autumn. The same goes for the MHS, especially when we observe the $UTCI_{21h}$. This finding is in line with the mentioned studies. Đorđević [91] analyzed the period from 1888 to 2006, and detected a notable positive temperature trend in Belgrade. The mean autumn temperature in Belgrade positive trend amounts to 0.74 °C/100 years, while for the minimum temperature the trend was 1.69 °C/100 years. This is one of the main reasons why the average autumn $UTCI_{14h}$ in Belgrade has recorded growth: during the first decade it was 15.71 °C, while during the second decade it was 16.30 °C (a difference of 0.59 °C). In Novi Sad, there is also an increase in the value of UTCIs, but it is somewhat slower and less pronounced. So, for example, if we compare the average decade-long value of $UTCI_{14h}$ in Novi Sad, the observed change is 0.22 °C, for $UTCI_{07h}$ it is 0.47 °C, and for $UTCI_{21h}$ it is 0.31 °C.

Under the permanent rise in global temperature, winters in this part of the world are becoming milder. During the winter, the most common days are those with NTS and SLCS. Winters are somewhat colder in Novi Sad, while in Belgrade and Niš they are slowly becoming more moderate. As Drljača et al. have underlined [92], due to the pronounced effect of UHI on the microclimate of the city, heat waves during the winter in Belgrade are not isolated events, but in terms of their intensity, they are not as extreme as the summer heat events. As Đorđević [91] claims, winters in Belgrade have become warmer. For instance, the mean winter temperature grows at the rate of 1.95 °C/100 years in this city. In the same way, the minimum winter temperature in Belgrade has a positive trend of 2.97 °C/100 years, which means that the rise of the minimum temperature was the greatest [91]. A similar situation prevails in the area of Novi Sad. According to Lazić et al. [90], the mean annual minimum temperature measured in Novi Sad (Rimski Šančevi) has shown a significant rising trend with 0.0108 °C/decade (over a period of 40 years). Such changes were also reflected in the urban OTC, which can be seen in Figure 5. Based on that, it is obvious that the share of winter days with NTS has increased over the years.

In the end, when we talk about OTC in urban areas, we must not forget the impact of anthropogenic factors including intensive urbanization, high population densities, industry, wide infrastructure corridors and commercial zones, domination of artificial materials used in construction (concrete, steel, asphalt, glass), combined with lack of open green spaces and street greenery, air pollution, etc. [10–16]. This leads to urban heat load which is manifested in higher air and surface temperatures compared to semi-natural surroundings [2,5,93,94]. The phenomenon of urban heat islands (UHI) is characteristic of large cities, especially those such as Belgrade, and in future research, the issue of OTC should receive attention. Andjelković [95] described Belgrade's UHI based on data collected from two meteorological stations (located in Vračar—the central part of the city, and in Surčin—a suburban area of Belgrade). Recently, Pecelj et al. made an important step forward in bioclimatic research of urban areas when they conducted a study that covered a period of 43 years, based on data from two weather stations located in different parts of Belgrade.

The greatest shortcoming and limitation of the methodology applied in this paper is actually that the research is based on data from only one meteorological station in each city. This way we cannot gain complete insight into the OTC in the entire urban area, but only in that part of the city where the synoptic stations are located. In the case of Belgrade and Niš, it is the central urban zone, while in Novi Sad it is the area between urban and suburban zone. A comprehensive OTC assessment can only be conducted using a more developed and concentrated network of urban and suburban meteorological measurements. This way, the overall effect of the previously mentioned anthropogenic factors that initiate the UHI

can be observed. The impact of topography, as well as green and blue infrastructure, can also be analyzed. Nonetheless, this study presents diverse and quality results that allow us to gain a general insight into the outdoor thermal comfort of these three cities during different seasons. Also, this study once again confirmed the applicability of UTCI in the study of urban bioclimatic conditions.

Given that Belgrade, Novi Sad, and Niš are in the process of intensive urban development, future studies should be linked to urban planning. Bioclimatic conditions and OTC should be considered as important instrument for sustainable urban planning. For example, they can play a significant role in the planning of city health zones, school zones and zones where social protection facilities are located. Furthermore, this also applies to the planning of industrial zones, central business zones, and other locations where employees spend a lot of time. Last but not least, this can be also important for housing, urban landscape planning, etc.

## 5. Conclusions

The main objective of this paper was the temporal assessment of OTC in three Serbian cities: Belgrade, Novi Sad, and Niš, during different seasons. An analysis of biothermal conditions was provided for three synoptic stations, and it covered a period of 20 years (from 1999 to 2018). Through the application of a heat budget index UTCI, seasonal and annual biothermal conditions were determined based on morning, midday and evening values (07 h, 14 h, and 21 h CET) of meteorological data.

Niš is rated as the city with the most pronounced thermal discomfort, especially during the summer season. The geographical position and morphological characteristics of the city significantly contribute to this. Novi Sad is identified as a city with generally more favorable OTC. The findings indicate a constant increase in the UTCI values. This results in the reduction in the number of days in all categories of cold stress during all seasons. Additionally, it causes an increase in the number of days when some of the categories of heat stress occur (MHS, SHS, VSHS, and EHS). UTCI seasonal anomalies are more prominent during the summer and spring, and somewhat less prominent during the winter. The average annual UTCI values also show a rising trend, with the increase ranging from 0.50 °C to 1.33 °C. The most significant deviations from the average UTCI values, both seasonal and annual, were observed during 2000, 2007, 2012, 2015, 2017, and 2018. A comparative analysis data from the two decades that were studied showed that after 2009, changes in the bioclimatic conditions of urban areas in Serbia accelerated, primarily in Belgrade and Niš.

The benefits of using the UTCI in urban thermal assessment were once again confirmed by this study. As stated at the beginning, this index is very accurate and it can be applied in different environments. The main limitation of the method applied here is that the study was based on data from only one meteorological station in each city. Consequently, the obtained results are somewhat limited, and cannot be applied to the entire urban area. As part of the future research, it is necessary to work on the development of a denser network of automatic micrometeorological measurement stations, based on which data from different parts of the city could be obtained. Only in this way it is possible to obtain a complete bioclimatic picture of the given urban spaces.

**Author Contributions:** Conceptualization and methodology, M.P.; writing—original draft preparation, M.L.; validation, M.L., D.F. and B.L.; formal analysis, M.L., L.C., L.D. and A.L.; data curation, A.L., L.C., A.V. and L.D.; writing—review and editing, M.P., M.L.; visualization, M.L., A.L., A.V. and L.D.; supervision, D.F., B.L. and M.P.; project administration, D.F., B.L. All authors participated in the analysis of the obtained results and made a significant contribution to the discussion chapter. All authors have read and agreed to the published version of the manuscript.

**Funding:** This research was funded by the Ministry of Education, Science and Technological Development (Republic of Serbia), grant number III 47007, 176017 and 176008.

**Institutional Review Board Statement:** Not applicable.

**Informed Consent Statement:** Not applicable.

**Data Availability Statement:** Not applicable.

**Acknowledgments:** The paper represents the results of research on the national projects supported by the Ministry of Education, Science and Technological Development, Republic of Serbia (nos. III 47007, 176017 and 176008).

**Conflicts of Interest:** The authors declare no conflict of interest.

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
