# Peer review of "Assessment of Outdoor Thermal Comfort in Serbia’s Urban Environments during Different Seasons"

_atmosphere, doi:10.3390/atmos12081084_

Round 1

Reviewer 1 Report

Please see the attached report

Reviewer 2 Report

Dear authors,

In my opinion your article is interesting and gives significant contribution to literature regarding human thermal comfort. Comparison of the three major cities in Serbia over a period of two decades is a good idea and gives the overview of the trends of thermal conditions in those urban areas.

I have one suggestion, and it is related to figures 2,3,4,5. Please insert the legend below each of them, so the reader don’t have to go up to the table 1 to check the colors.

Kind regards,

Author Response

Based on the Reviewer's suggestion, the legend, ie. color-bar for each figure is added.

Reviewer 3 Report

This paper investigates the outdoor thermal comfort in three urban environments, however, some issues should be addressed before considering for publication in the Journal of building engineering.

Research gap: The authors have provided a literature review in section 1 but the research that is referenced in section 1 are not sufficient to show the research gaps and contribution of the research. I suggest that authors revise the introduction section to highlight their contribution to research. Authors can use recently published papers to highlight the gaps in the existing body of the literature.

At the end of the introduction section, the authors should present the objectives and goals of their study. They should highlight that how the identified gaps would be addressed in this paper or they can highlight their research questions. In this version, the research question is unclear.

Authors should have a methodology section that clearly states the steps of their work and the input/outputs of each step. The methodology should address the gaps identified in the previous section. Now in the version, the Materials and Method section is not clear enough. It is more similar to the literature review and lacks necessary information about the methodology. In this section, the authors should clarify the tools that they have used in their research. How these tools are calibrated and applied according to the research methodology? The other point that should be highlighted in the section is that what is the validation process in this research, in this way the results would be reliable.

The discussion section is not deep enough and is similar to the results section. I recommend that authors include more insights/benefits of their work bases on the proposed solutions in the previous sections. Then a summary of these insights can be highlighted in the conclusion section again.

The other point is about the generalization of their prosed methodology. They should highlight that which parts of their methodology is case-specific (scale of study or climate or, …) and which part is generalizable.

In the conclusion, the authors should mention the limitation of their methodology as a suggestion for future research. I recommend authors rewrite the conclusion based on the above points.

Round 2

Reviewer 1 Report

Authors have worked on all the issues I highlighted in my first review report. I believe that this manuscript is almost ready for publication in Atmosphere. However the manuscript needs further English editing because it contains grammatical/typo mistakes. For example line 101: This section consist should be This section consists, Line 128 This part if the city should be This part of the city etc.

I hope, if this manuscript is finally accepted, that these issues is resolved even before the production step.

Author Response

Based on your advice, we have hired a qualified English-speaking Editor with wide experience in professional editing of various articles.

We are sending you a version with corrected grammatical/typo mistakes, which proves that the paper has been edited.

Reviewer 2 Report

None

Author Response

Based on the advice of other Reviewers, we have hired a professional English-speaking Editor. The Editor performed all the necessary checks and corrections, especially Language (sentence construction, English word choice, clarity, tone, voice, etc.) and Grammar checks.   

As proof of that, we are sending you a version of the article with all corrections (marked in red).
